

# SPREADS: From Research to Operational Open-Source Data Assimilation System

Carla Cardinali[1]

Giovanni Conti[1]

Marcelo Guatura[1]

Sami Saarinen[2]

Luis Gustavo Gonçalves De Gonçalves[1]

Jeffrey Anderson[3]

Kevin Raeder[3]

1- CMCC-Foundation Euro-Mediterranean Center on Climate Change, Italy
2-HPC-consulting Ltd, Finland
3- NSF National Center for Atmospheric Research (NSF-NCAR), Boulder, Colorado, USA

Corresponding author: Carla Cardinali, carla.cardinali@cmcc.it



24          **Abstract**

The Scalable PaRallelised EArth Data Assimilation System (SPREADS) is a next-

generation  data  assimilation  system  developed  at  CMCC-Foundation  (Euro-
Mediterranean Center on Climate Change) to support operational global forecasts.
Built upon the Data Assimilation Research Testbed (DART), SPREADS incorporates
key  advancements  such  as  First  Guess  at  Appropriate  Time  (FGAT),  enhanced
observation handling via D4O (Database for Observations), and high-performance
parallelisation to significantly improve computational efficiency and scalability. A
major  focus  of  SPREADS  is  the  assimilation  of  a  vastly  increased  number  of
asynchronous  satellite-based  radiances,  which  have  been  shown  to  substantially
enhance  analysis  quality.  Designed  for  coupled  atmosphere-land-ocean-ice
assimilation, SPREADS forms the core of the CMCC Earth SYstem Modelling and Data
Assimilation (ESYDA division) operational forecast system. This paper presents the
modifications  made  to  DART,  evaluates  preliminary  results,  and  outlines  future
developments toward fully coupled data assimilation.

**1.  Introduction & Motivation**

Data  Assimilation  Research  Testbed  (DART;  The  Data  Assimilation  Research

Testbed, Version X.Y.Z, 2021, Boulder, Colorado UCAR/NSF NCAR/CISL/DAReS,
http://doi.org/10.5065/D6WQ0202), is a widely used data assimilation system in the
atmospheric and oceanic sciences. It was developed and is maintained by the Data
Assimilation Research Section (DAReS) at the NSF National Center for Atmospheric
Research (NCAR) in the United States. DART provides a flexible and modular platform
for  conducting  research  on  data  assimilation  algorithms  and  their  applications  to
numerical  weather  prediction,  climate  modelling,  and  other  environmental
forecasting  systems.  It  primarily  focuses  on  ensemble-based  data  assimilation
methods, such as the Ensemble Kalman Filter (EnKF) and its variants (Evensen
1994a,b; Evensen 2001; Tippett *et al.*, 2003; Collins 2007). These methods use an
ensemble of model state vectors to represent the uncertainty in the system state and
assimilate observations to update this ensemble. DART is designed with a modular
architecture,  allowing  users  to  easily  integrate  different  numerical  models,





observation types, data assimilation algorithms, and experimental configurations.
This flexibility enables researchers to tailor the system to specific research questions
and applications. DART incorporates the localisation technique (Hamill *et al.* 2001;
Houtekamer and Mitchell 2001; Haugen and Evensen 2002 Otto *et al*. 2004) to account
for the fact that observations are often only informative within a limited spatial and
temporal range, helping therefore to prevent spurious long-range correlations in the
analysis, improving the accuracy of the assimilation. DART  also includes methods for
adaptively inflating the ensemble spread to account for underestimation or
overestimation of forecast error covariance. Adaptive inflation is crucial for
maintaining the reliability of the ensemble and preventing filter divergence. The
benefit to the scientific community of using DART all over these years is without any
doubts: DART has been used by a large world wide young and senior researchers to
advance understanding data assimilation methods and observations usage (Noh *et al.*
2024; Tang *et al.* 2024; Dietrich *et al.* 2024; Pedatella and Anderson 2022; Fox *et al*
2022; Rackza *et al.* 2021). Over the last 20 years, DART has been continuously
developed and improved with input and feedback from the scientific community
across the atmospheric, oceanic and land sciences. Researchers have contributed new
algorithms, techniques, and methodologies expanding the capabilities of the
framework and  enabling new users to experiment with state-of-the-art approaches
(Grooms and  Riedel 2024; Anderson 2023; Dibia 2023). By experimenting with
various observation types, processing techniques, and quality control methods, users
have contributed to optimising the assimilation of observational data and the
computational efficiency of DART. This includes improvements in parallelisation
strategies, algorithmic optimisations, and enhancements to reduce memory usage and
computational costs.  And finally also improvements have been obtained on
diagnostics, metrics, and benchmarking datasets to assess the quality and reliability
of the produced analyses.
In 2021 the Euro-Mediterranean Center on Climate Change (CMCC, Italy)
approved a new strategy on longer forecast range predictions (e.g. the seasonal
forecast) strongly supporting the use of a proper initialisation of these predictions by
a weakly coupled data assimilation system. The CMCC strategy foresaw therefore the
development of a weakly coupled atmosphere, land, ocean and cryosphere data
assimilation system initialising such predictions. Given the crucial role that the open-



source modelling plays in terms of transparency and reproducibility, fostering
collaboration and community engagement that encourages knowledge sharing, idea
exchange, and collective problem-solving, leading to the development of more robust
and comprehensive models, CMCC has engaged in the development of a data
assimilation system that will serve as an open-source system for operational use.
Therefore starting from the open-source DART, SPREADS has evolved by modifying
and implementing new features essential for an operational use of the system
(Cardinali *et al.*, 2025). CMCC's development of SPREADS as an open-source data
assimilation system for operational use builds upon the strengths of the DART
framework while customising and extending it to meet the specific requirements of
operational forecasting and decision support. In this paper, the description of the
changes adopted towards an operational use of an atmospheric data assimilation
system is described and assessed. This paper describes the methodological
innovations in SPREADS, evaluates its preliminary performance, and outlines future
expansion plans toward a coupled open-source DA system.

**2.  Ensemble Kalman Filter and SPREADS**

The Ensemble Adjustment Kalman Filter (EAKF) is a data assimilation technique
developed by members of the DAReS team  (Anderson 2001, 2003; Andersson  2009;
Andersson, 2012; Anderson and Collins 2012; Reader et al. 2012) within DART. The
EAKF addresses key limitations of the standard EnKF, particularly when dealing with
small ensemble sizes or poorly known model and observation error statistics. It
employs a least squares method to adjust the ensemble state, ensuring consistency
with both model dynamics and observational constraints. The EAKF refines the
ensemble mean and spread to better fit incoming observations. Observations and
ensemble members are assimilated within localised regions, reducing the impact of
spurious long-range correlations and enhancing computational efficiency.
When new observations become available, the EAKF assimilation process follows
two main steps:
1. Observation-Space Update (Scalar Update): for each observation, the

ensemble members are first updated in the observation space. This involves

adjusting the prior observation estimates for each ensemble member based on the





observed value, the prior ensemble's variance in observation space, and the
observation error variance. This step ensures that the updated observation
estimates are consistent with the new measurement. This update is often
performed as a series of scalar updates if observations are assimilated
sequentially.
2. State-Space Adjustment (Ensemble Member Transformation): following the
observation-space update, each ensemble member's state variables are then
adjusted to reflect the change made in the observation space, ensuring that the
updated state remains consistent with the updated observation and the ensemble's
internal correlations. This adjustment explicitly leverages the cross-
covariances between each state variable and the observed variable (computed
directly from the prior ensemble). The transformation ensures that the ensemble
mean and covariance are updated according to the Kalman filter equations, and
crucially, that the updated ensemble members retain their statistical spread and
do not collapse. This adjustment is applied to each ensemble member
independently, guaranteeing that the analysis ensemble still represents the
posterior uncertainty.
By explicitly incorporating the least squares assumption, the EAKF provides a
computationally efficient solution. Under these assumptions, the ensemble filtering
problem reduces to a nonlinear filter applied to a scalar, followed by sequential linear
regressions. While subsets of observations with independent error distributions can
be assimilated in sequence, the sequential nature of the regression step presents a
computational challenge when millions of observations must be processed within a
six-hour window. It is well know that satellite radiance assimilation has significantly
improved the quality of numerical weather prediction (NWP) analyses: incorporating
vast amounts of satellite observations, it leads to better initial conditions for
forecasting. To efficiently handle the assimilation of large volumes of satellite
radiances, SPREADS has introduced several modifications, including the First Guess at
Appropriate Time (FGAT) approach, the implementation of RTTOV (the Radiative
Transfer for TIROS Operational Vertical Sounder, Saunders et al., 2018) for radiance
processing (Kugler et al. 2023), the scan and air mass bias corrections, and code
improvements in the observations treatment.



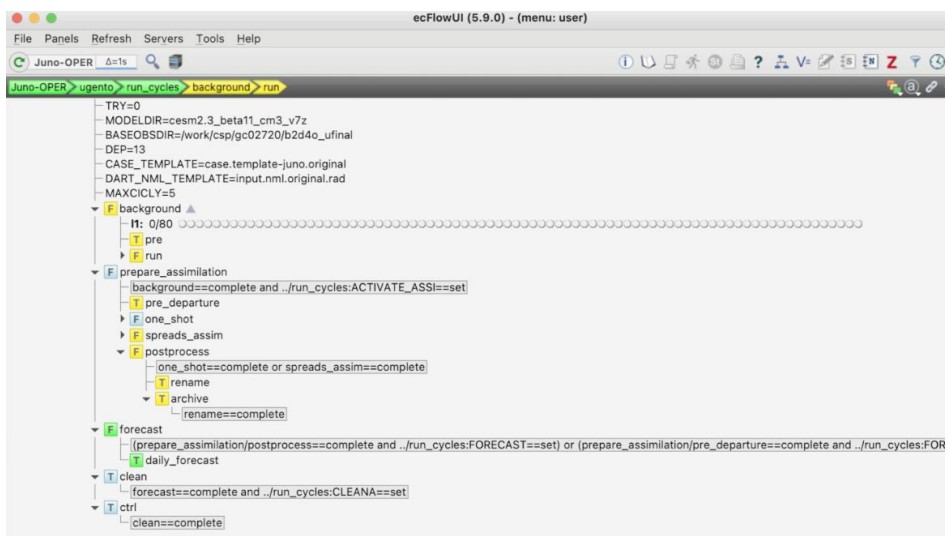

*Figure 1: Graphical User Interface EcFlow managing the atmosphere and land component data assimilation and forecast of SPREADS*

### 2.1 Graphical User Interface and Diagnostics

To support the execution of SPREADS, an ensemble data assimilation system processing over one million observations every six hours, a dedicated Graphical User Interface (GUI) was essential. Given the complexity of SPREADS, which involves numerous interdependent modules and programs, a robust and dynamic workflow management tool was required. For this reason, an EcFlow-based GUI (ECMWF EcFlow User Documentation https://confluence.ecmwf.int/display/ECFLOW) was developed in parallel with SPREADS. This client/server interface allows for controlled and coordinated execution of all components of the data assimilation suite. The GUI handles task scheduling, monitors job statuses, and responds to events via embedded script commands. It is designed to tolerate hardware or software failures and supports automatic restarts when needed. The GUI manages task dependencies using a trigger-based system, where the execution of one job can depend on the status of others. Job statuses typically include submission, queuing, running, failed, or suspended. These functionalities are enabled by a combination of command-line executables, shared libraries, and a Python-based interface that defines the suite structure and handles communication with the EcFlow server. The server component acts as the scheduler, responding to client requests and managing job execution. While





not a queuing system itself, it is capable of submitting jobs to external queueing
systems, making it suitable for heterogeneous computing environments. The GUI also
provides real-time monitoring and visualisation of the suite's hierarchical node tree,
giving users full visibility and control over the operational workflow. Figure 1 shows
the EcFlow GUI for SPREADS.
Alongside the EcFlow GUI, an interactive graphical diagnostics package based on
Streamlit (Streamlit 2024) has been developed to monitor and assess the performance
of SPREADS. This tool allows users to explore both model and observation spaces
through a range of visualisations. In model space, users can generate geographical
maps, cross sections, and mean vertical profiles of key variables. In observation space,
the package offers time series and vertical profiles of key statistics, such as biases and
standard deviations of the differences between observations and both the prior and
posterior fields. Additional diagnostic and dynamical metrics are also available. All the
plots presented in this paper were produced using this diagnostic tool.

*2.2 FGAT-approach & Code Modularity*
One of the key features developed in SPREADS is the FGAT approach, designed to
enhance both efficiency and accuracy in the assimilation process. In the First Guess at
Appropriate Time (FGAT) approach, the model's background (first-guess) forecast is
interpolated to the exact time of each observation, so that the observation-minus-
background difference is evaluated at the proper moment before the analysis is
performed, yielding a more consistent estimate of the system's current state.
In SPREADS, the model's first guess is interpolated to match the time of
observational data. To achieve this, the commonly used 6-hour assimilation window
is divided into 11 time slots of 30 minutes and 2 additional slots of 15 minutes at the
beginning and end of the window. This fine-grained temporal segmentation ensures
precise alignment between observations and model output, enabling a more
meaningful and accurate comparison.
This approach offers several key advantages: it eliminates the need to time-shift
either field, which (a) removes interpolation-induced errors and (b) saves
considerable CPU and memory, because one forecast integration serves all
observation times instead of many separate. Also it enhances consistency by ensuring



alignment between the model's initial conditions and observational data, enabling
therefore the assimilation of asynchronous observations such as polar orbiting
satellite-based observations. Finally, since FGAT can be applied across various data
assimilation techniques, it provides flexibility in adapting to different modelling and
observational setups.
The effectiveness of FGAT was demonstrated during the development of the
variational data assimilation system at ECMWF, where the transition from a 6-hour
3DVar window to FGAT-3DVar resulted in the largest improvement in assimilation
performance (Andersson et al., 1998). The modularity inherent in DART has been
further enhanced in SPREADS by refining and structuring the assimilation steps into
four distinct modules, each designed to improve efficiency, computational
organisation and flexibility. Notably, these modules can be executed independently,
allowing for greater adaptability in different assimilation workflows.
*Module-0:* Executes the model trajectory using the FGAT approach, while
independently handling observation preprocessing.
*Module-1*: Performs observation preprocessing, including cross-checking
observations, converting them from buffer storage to an SQL query-based database,
and conducting screening and blacklisting. Also the scan and air mass bias corrections
are performed in here: biases in satellite radiance observations arise due to
instrument calibration errors, radiative transfer modeling inaccuracies, and
atmospheric variations. The implementation of bias corrections includes adjusting
radiances for systematic errors associated with the sensor's viewing angle and using
predictors such as atmospheric thickness and surface temperature to adjust radiances
for biases linked to atmospheric conditions (Harris and Kelly, 2001, Auligné *et al.* ,
2007).

*Module-2*: Carries out the nonlinear spatial interpolation of model values from all
ensemble members to the observation locations.
*Module-3*: Executes the two-step sequential regression, ensuring the adjustment
of the model state based on observations.
Since each module can be executed independently, users can customise the
assimilation process by running only the required components, optimising
computational resources and allowing for seamless integration with external systems.





This flexibility enhances scalability, maintainability, and operational efficiency,
making SPREADS highly adaptable for various forecasting and research applications.
*2.3 Observation handling d4o*
To ensure a flexible and fast observation handling throughout all the assimilation
processes a query language observation database (Database for Observations, d4o)
based on SQLite (https://sqlite.org/) open-source has been developed. D4o manages
and controls the observations flow through observations definition and query
language organised in a hierarchical tree-like structure from which is easy to select
the desired information and place it in a data matrix for further examination. The
system allows SQL queries to efficiently extract and manipulate observational data.
The provided SQL can for example filter observations based on latitude, observation
type, quality control flags, and availability of posterior values. A Fortran module
(fd4o_mod) was developed to interface with the database. It includes functions to
open, close, query, and update the database. The Fortran interface is built on a C-layer
that calls standard SQLite APIs. The d4o database was integrated into the SPREADS
data assimilation system to handle large volumes of observational data. This
implementation optimises I/O operations and improves data accessibility across
parallel processing tasks. In fact, this relational-like process is particularly efficient for
MPI-parallel data access and queries coordination for data shuffling between MPI-
tasks. High vectorisation efficiency for storing and retrieving observations is therefore
achieved, enabling fast, flexible and configurable I/O management.
The Fortran interface was optimised for MPI-parallelised operations: the
observations are now stored and retrieved in a highly efficient, vectorised manner to
reduce computational overhead. Various parallelisation techniques, such as OpenMP
and MPI non-blocking communications and several debugging and logging options
were introduced to track database transactions. The database system was extended
to support observational inputs for different Earth system models, including CAM
(Community Atmosphere Model) and CLM (Community Land Model).
The d4o database has been architected to handle, in a fully scalable manner, the
vast volumes of satellite observations required for state-of-the-art analyses. It draws
its observation metadata directly from a series of database files ("data pools"), whose
contents are cached into the observational I/O-server tasks. The design of d4o





deliberately builds on the success of the ECMWF ODB system (Observational
DataBase) first deployed in 2000 to seamlessly assimilate diverse observation types
and, in particular, to manage very large volumes of IASI radiance data within the
ECMWF          IFS          4DVar          framework
(https://www.ecmwf.int/sites/default/files/elibrary/2004/76278-ifs-
documentation-cy36r1-part-i-observation-processing_1.pdf). In contrast to ODB, d4o
leverages a modern, standardised SQL-query interface via the lightweight SQLite
engine (https://www.sqlite.org/) wrapped in a versatile Fortran 2008 SPREADS
interface that employs hybrid MPI/OpenMP parallelism and parallel I/O for the SQLite
files. This hybrid approach reduces total memory usage and cuts the need for large
MPI task counts, avoiding the overhead of fine-grained message passing and
synchronisation: so that, in practice, 2–8 threads per MPI task suffice. Indeed, on Sami
Saarinen's personal tests, SPREADS with its optimised d4o library running on just
eight nodes outperformed a comparable DART run on 32 nodes by a factor of 2–3
(2024–2025, pers. comm.).

Module-1 provides all the observations preprocessing: once all the observations
to be assimilated are in d4o, the screening according to the chosen resolution is
performed and the observations are thinned accordingly. The screening module was
implemented to filter out low-quality or irrelevant observations.

Moreover, in SPREADS, a dedicated blacklisting module has been introduced that
permanently excludes observation channels, platforms, or stations with documented
systematic errors at ingest, preventing the assimilation of problematic data and
reducing subsequent quality-control and computational overhead.

*2.4 Code optimisation*

The modifications introduced focused on optimising code efficiency and
enhancing parallelisation. The traditional linked-list observation sequence was
replaced with a SQLite-based d4o database system, significantly improving memory
management and data retrieval speed. An I/O server architecture was implemented
to separate data handling from computational tasks, reducing bottlenecks and
improving scalability. The sequential observation loop in Module-3 was optimised by
eliminating unnecessary index copying and improving observation-state closeness
calculations, leading to faster processing. Parallelisation was enhanced through the



introduction of OpenMP within MPI tasks, reducing the number of required MPI tasks
while maximising computational efficiency.
The previous blocking MPI communications were replaced with non-blocking
alternatives to minimise delays and improve data exchange. Additionally, database
operations were optimised by disabling SQLite journaling, allowing for faster
database writes and exclusive access for I/O servers. The handling of satellite
observations, particularly IASI data, was refined with a more efficient preprocessing
pipeline. Performance monitoring tools, such as perfstat, were introduced to identify
and address bottlenecks, while automated scripts were developed to streamline
database management, observation blacklisting, and debugging. These modifications
collectively enhanced the scalability, performance, and reliability of the SPREADS data
assimilation system. Table 1 shows the computational speed and efficiency before and
after the code optimisation.

| Module | CPU | | Node | | Member | |
|---|---|---|---|---|---|---|
| | old | new | old | new | old | new |
| Module0 | | | | | | |
| Module1 | 105' | 38' | 2 | 1 | 1 | 1 |
| Module2 | 8' | 6' | 12 x ts | 4 x ts | | |
| Module3 | 1ʰ30 | 35' | 25 | 8 | | |

*Table 1: Computer configuration and CPU time before (left panel) and after (right panel) SPREADS optimisation. Ts stands for time-slot.*


**3. E-suite and preliminary results**
SPREADS is fully integrated into the Community Earth System Model (CESM,
https://www.cesm.ucar.edu/), an infrastructure developed through a joint
collaboration between many meteorological centres including CMCC








and NCAR. CESM provides a flexible software framework for configuring and

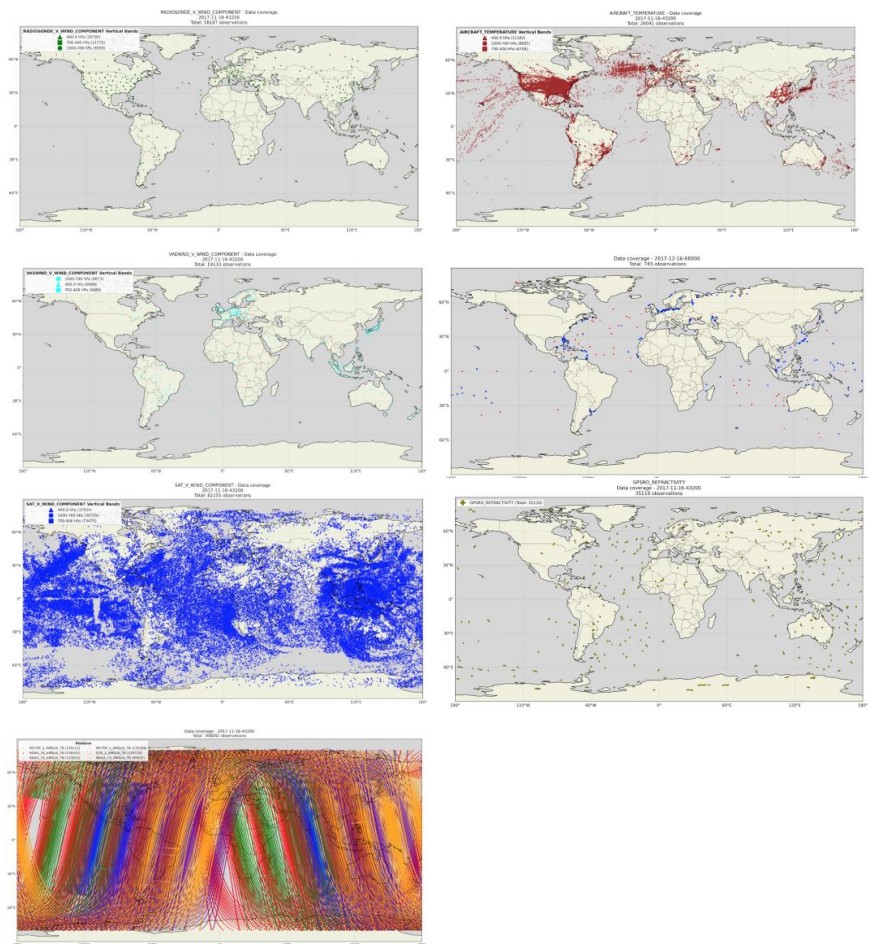

*Figure 2: 6 hour window observation data coverage according to observation type. From top left: Radiosonde, Aircraft, Wind profiler, Synop and Buoy, AMV, GPS-RO and AMSU-A*

running coupled models, each designed to represent different components of the
Earth system. Specifically, SPREADS is coupled with the atmosphere model (CAM), the
land model (CLM), the cryosphere model (CICE), and the ocean model NEMO (for the
CMCC CESM).





| Observation | Observation kind | Information |
|---|---|---|
| GPS-RO Metop A, B GRAS | Refractivity | Temperature |
| AMSU-A Metop A | Microwave sounder radiance | Temperature |
| AMSU-A Metop B | | |
| AMSU-A NOAA-15 | | |
| AMSU-A NOAA 18 | | |
| AMSU-A NOAA-19 | | |
| AMSU-A AQUA | | |
| AMV AQUA | Visible | u, v |
| AMV TERRA | Visible | |
| AMV GOES-15 | Visible | |
| AMV Meteosat-10 | IR, WV and V | |
| AMV COMS-1 | IR, WV and V | |
| AMV Dual Metop | IR | |
| AMV INSAT -3D | | |
| AMV NPP | IR | |
| AMV HIMAWARI-8 | Visible | |
| Profiler | European, Japanese Wind | u, v |
| Radiosonde | Land and Ship | u, v,  T, q |
| Aircraft | | u, v, T, q |
| Buoys | Moored and Drifters | Surface pressure |
| SYNOP | Land | Surface pressure |

*Table 2 Observation types and platforms assimilated*


An experimental suite (E-suite) has been implemented using SPREADS to assess
its performance in an operational-like environment. The E-suite began running in
January 2024, covering the period from July 2017 to the present at 1° horizontal
resolution. It is currently up to the January 2018 analysis production, utilising the 93
model levels of CAM (version 6 finite volume dynamical core; Simpson *et al,* 2025),
with enhanced vertical resolution in the free troposphere and stratosphere and a



model lid height set at 0.01 hPa. Each cycle operates on a 6-hour window, subdivided
via the FGAT approach to accommodate the asynchronous nature of the observations.
The E-suite assimilates a wide range of observations, as listed in Table 2.
A representative data coverage for a 6-hour assimilation window centred on 12
UTC is shown in Figure 2. For that cycle, the number of assimilated observations

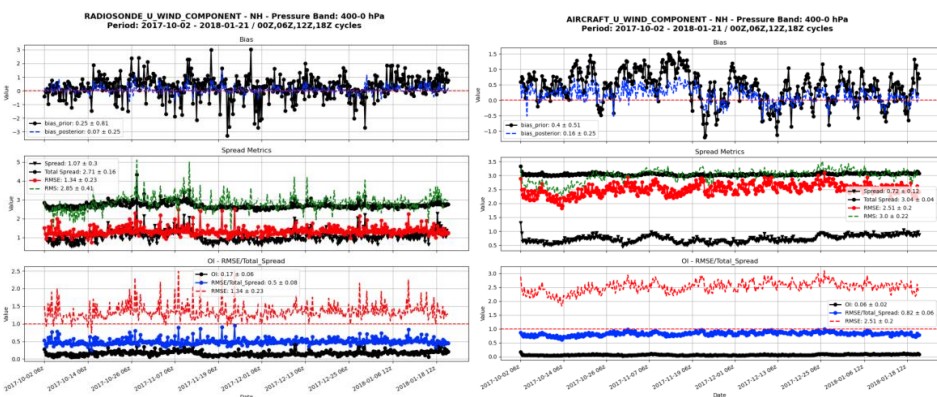

*Figure 3: Time series of observation-minus-background (prior) and observation-minus-analysis (posterior) departures for the zonal wind component (u) from 2 October 2017 to 21 January 2018 at 00, 06, 12, and 18 UTC, over the Northern Hemisphere and between 400 hPa and 0.01 hPa. The left panel shows radiosonde observations, and the right panel shows aircraft observations. Displayed diagnostics include RMS, RMSE, ensemble Dispersion Index (DI), and Observation Influence (OI*

includes: 35,110 GPS-RO refractivity profiles, 768,002 AMSU-A brightness
temperatures, 163,284 AMV winds, 40,158 wind profiler reports, 174,263 radiosonde
measurements, 79,043 aircraft reports, and 500 surface pressure observations from
SYNOP and BUOY platforms. This results in a total of approximately $1.3 \times 10^6$
assimilated observations every 6 hours. Figure 3 presents time series of the zonal
wind component (u) from radiosonde (left) and aircraft (right) observations over the
Northern Hemisphere between 400 hPa and 0.01 hPa. The assimilation results show
a reduction in bias of approximately 30% for radiosondes and 40% for aircraft data.
Aircraft observations typically exhibit smaller biases, generally within ±1.5 m/s,
compared to radiosonde data, which show biases reaching ±3 m/s.





356 A notable change occurs around 14 November 2017, coinciding with the
357 introduction of AMSU-A microwave radiances into the assimilation system; prior to
358 this, aircraft data tend to underestimate the zonal wind.

359 The RMSE for aircraft observations remains higher (~2.5 m/s) than that for
360 radiosondes (~1.5 m/s), consistent with known instrument characteristics and
361 sampling differences. The ensemble Dispersion Index (DI), defined as the ratio of the
362 ensemble RMSE (computed against the own analysis) to the total spread, stays close
363 to 1 throughout the period for the aircraft observation types, whilst is less than 1 for
364 the radiosonde observations indicating an overdispersive ensemble at the top of the
365 atmosphere (~50hPa) and a well calibrated ensemble in the high troposphere (~250
366 hPa).

367 The observation Influence (OI), $0 \leq OI \leq 1$, indicates that when OI = 0 the
368 observation had no leverage in the fit, whereas OI = 1 means the fit relied entirely on
369 the observation, with no contribution from the first guess (Cardinali *et al.*, 2004;
370 Cardinali, 2014; Liu *et al.*, 2009; Gharamti *et al.*, 2019). OI remains relatively low for
371 aircraft (~0.1), while radiosondes display higher influence values (~0.25), reflecting
372 their higher information content and smaller observation error variances in the upper
373 troposphere and lower stratosphere.

374 AMSU-A channels 9–14 were assimilated starting on 14 November 2017, initially
375 with a scan bias correction following Harris and Kelly (2001) and subsequently, on 2
376 January 2018, with the addition of an air-mass bias correction following Noh *et al.*
377 (2023). While the scan bias correction, after extensive evaluation, was shown to
378 perform satisfactorily, the air-mass bias correction proved less effective.

379 Further analysis demonstrated that the regression predictors used for the 200–
380 50 hPa thickness were inadequate to represent channels 11–14. A more suitable
381 choice was to separate the predictors and also include a 50–2 hPa thickness to account
382 for the stratospheric channels. Therefore, the final air-mass bias correction was
383 applied using a linear combination of several thickness predictors (1000–300, 200–
384 50, and 50–2 hPa), enabling the scheme to account for biases arising from multiple
385 physical dependencies simultaneously (Auligné *et al.*, 2007). To anchor the model



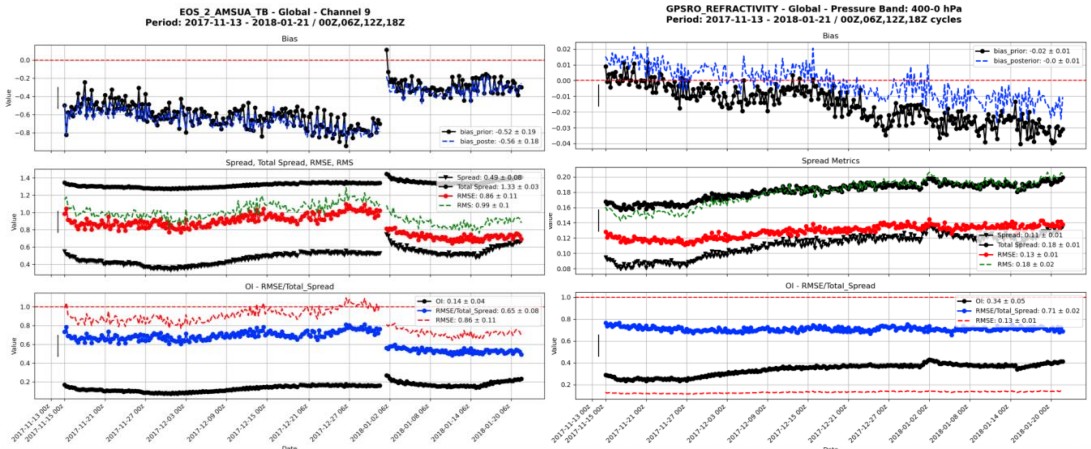

Figure 4: Global time series of Aqua AMSU-A TB channel 9 (left panel) compared with GPS-RO refractivity in the layer 400-0.01 hPa (right panel).

bias, unbiased radiosonde and GPS-RO observations were used, and AMSU-A channel
14 was left uncorrected since it is considered unbiased for the same reason. In
addition, the averaging of correction coefficients across last four cycles, as done in Noh
*et al.*, was found to mask the actual dynamical situation. To better capture flow-
dependent variability, our implementation used only the predictors within the current
assimilation window for the regression.
Figure 4 presents the time series diagnostics for two key observation types, GPS-
RO refractivity (right) and AMSU-A AQUA Channel 9 brightness temperatures (left),
covering the global pressure range from 400 to 0 hPa over the period 20171113 to
20180122. The GPS-RO diagnostics indicate a well-calibrated assimilation system: the
posterior bias remains near zero (0.00 ±0.01), the RMS and total spread are closely
matched, and the dispersion index remains near one. The observation influence (OI)
is moderate (0.34 ± 0.05), showing a balanced contribution between the observations
and the model background. These results confirm GPS-RO's role as a high-impact,
high-precision observation source, especially in the upper troposphere and
stratosphere.
In contrast, AMSU-A Channel 9, which peaks near 100 hPa in the mid-
stratosphere, presents a more complex picture. Following the introduction of scan
angle bias correction on 14 November and air-mass bias correction on 2 January, a



notable reduction in posterior bias is observed from ~0.6 K to -0.2 K. This
improvement in bias is accompanied by a steady decrease in RMSE from ~1 to 0.8 K.
Additionally, the observation influence rises to a higher value of 0.25, indicating a
better leveraged analysis fit.

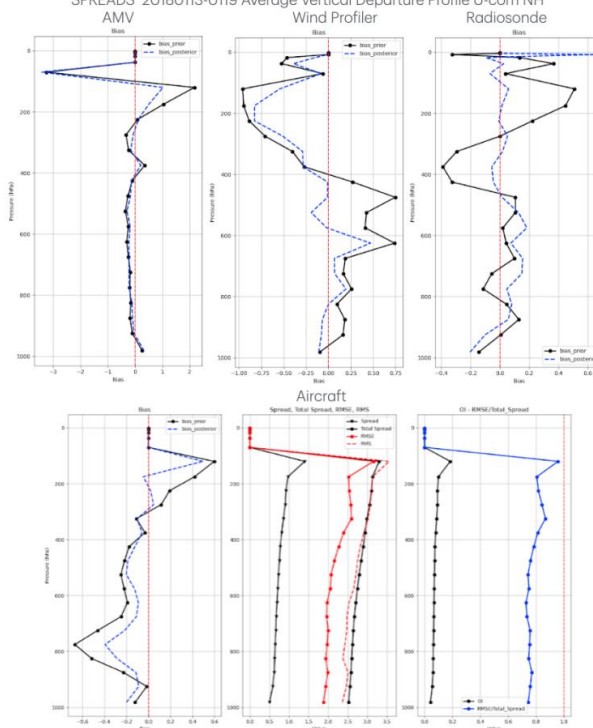

*Figure 5: Average vertical profiles of the zonal wind component (u) for AMVs, wind profilers, and radiosondes (top panel), and for aircraft observations (bottom panel). Aircraft profiles are complemented by diagnostics including RMS, RMSE, Dispersion Index (DI), and Observation Influence (OI).*

This diagnostic reinforces conclusions that while SPREADS effectively assimilates
high-accuracy GPS-RO data, stratospheric radiance assimilation still presents some
challenges.
Figure 5 presents a comparison of the prior and posterior departure vertical
profiles of the zonal wind, averaged over the week 20180103–19, for AMV, Wind
Profiler, and Radiosonde observations (top panel), and Aircraft observations (bottom





panel). For the Aircraft data, ensemble performance statistics are included: ensemble
spread, total spread, RMS, RMSE, dispersion index, and observation influence (OI).
Across all observation types, the posterior fit demonstrates a clear reduction in
bias throughout the atmosphere. Larger residual departures are seen above 200 hPa,
with AMVs showing differences between −3 m/s at 50 hPa and +1 m/s at 100 hPa.
Wind Profiler data shows a −0.75 m/s departure at 200 hPa, while Aircraft exhibit
slightly positive departures of 0.4 m/s at around 100 hPa. These discrepancies likely
stem from the sparser data coverage at upper levels and some residual effects from
AMSU-A assimilation.
The profiles also show that posterior fits tend to converge across observing
systems in the troposphere, suggesting a consistent adjustment by the assimilation
system despite differing data characteristics. A subtle transition in departure
behaviour is visible near the tropopause, possibly indicating increased
representativeness error or limitations in vertical resolution at this level.

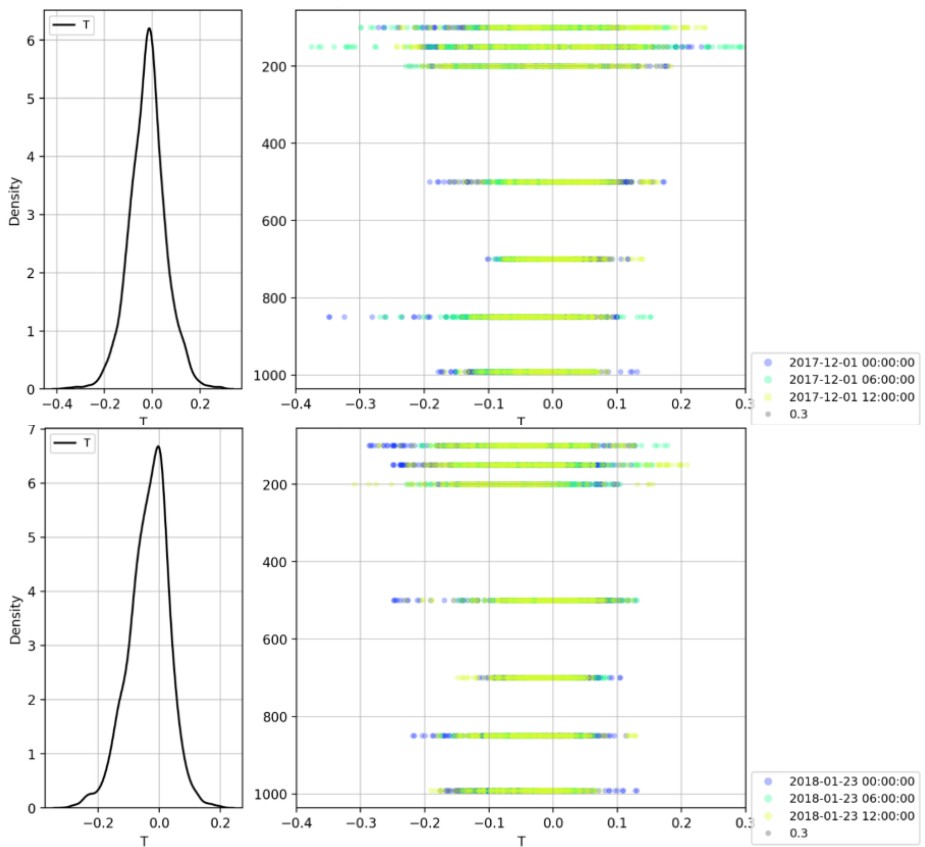

*Figure 6: Temperature increment density distribution for 20171201 (top panel) and 20180123 (bottom panel). Probability density function of the increments (left) and their vertical distribution across 3 cycles (right)*



Ensemble performance metrics are shown only for Aircraft data, as their
behaviour is representative of the other platforms. The RMSE remains around 2 m/s
throughout the column, while the RMS departures (middle panel, bottom row) are
slightly higher, as expected. The dispersion index indicates approximately 20% over-
dispersion in the ensemble below 200 hPa. The OI for Aircraft observations is
relatively low (0.1 to 0.2), primarily due to their error characteristics.
Finally, since the statistics are averaged over a full week, they represent
systematic patterns rather than short-term variability. Interpretation of vertical
features should also consider the varying vertical resolution and density of each
observation type, particularly above 200 hPa, where reduced data availability can
influence both bias correction and ensemble reliability.
Figure 6 illustrates the temperature increment density distribution for two
selected dates during the analysis period: December 1st, 2017 (top panel) and January
23rd, 2018 (bottom panel). Each panel shows the probability density function of the
increments (left) and their vertical distribution across three analysis cycles of each
day (right).
Over the course of the period, a clear reduction in the amplitude of temperature
increments is observed, particularly within the troposphere. The spread of the
distribution narrows from approximately ±0.4°K on December 1st to about ±0.2°K by
January 23rd. This contraction reflects improved constraint in the analysis, likely
resulting from better-calibrated observations and/or enhanced ensemble
performance.
Alongside the amplitude reduction, the tropospheric increment distributions
become increasingly symmetric around zero, indicating a progressive reduction in
systematic bias and a more balanced assimilation system. Additionally, the analysis
cycles per day show greater consistency over time: while 20171201 displays notable
variability between cycles, especially in the lower troposphere, the cycles on
20180123 exhibit much tighter agreement, suggesting improved temporal stability of
the system. Similar reduction of the increments is observed in the stratosphere.





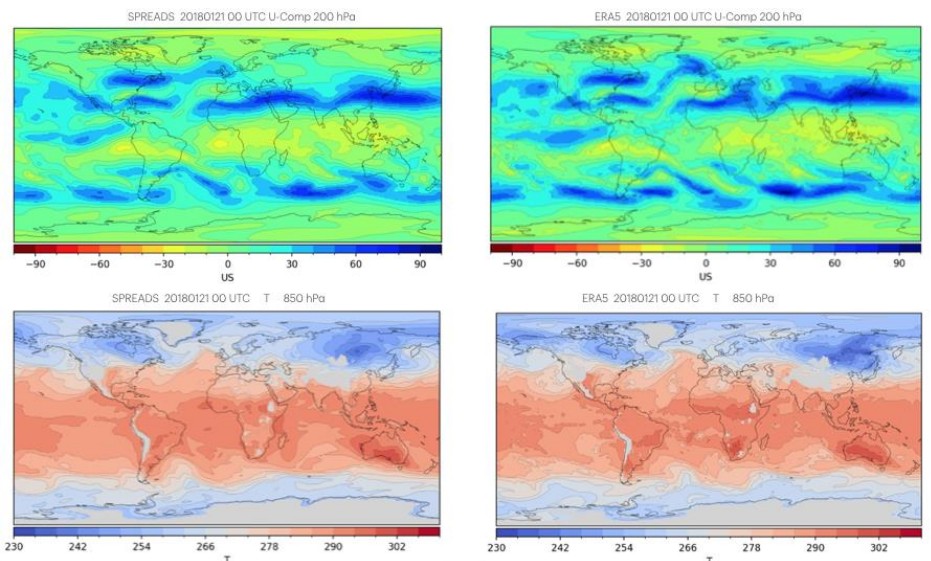

*Figure 7: SPREADS analysis (left panels) compare with ERA5 (right panels) of the U wind component at 200 hPa (top panels) and T at 850 hPa (bottom panels)*

Figures 7 and 8 present a comparison between SPREADS and ERA5 analyses,
highlighting both horizontal and vertical structural differences in the representation
of key atmospheric variables.

Figure 7 displays global fields from SPREADS (left panels) and ERA5 (right
panels) at 00 UTC on 20180121 for the zonal wind component at 200 hPa (top) and
temperature at 850 hPa (bottom). The large-scale circulation patterns are well
captured in SPREADS, showing good agreement with ERA5 in both magnitude and
spatial structure. However, differences emerge at smaller spatial scales, particularly
in regions with sharp gradients such as subtropical jet streams, where ERA5 exhibits
finer, more coherent jet streaks owing to its higher horizontal resolution nearly
double that of SPREADS.

At 850 hPa, the temperature patterns in both systems reflect realistic meridional
gradients and contrast between land and ocean. Yet, subtle regional differences are
visible: SPREADS appears cooler over high-latitude



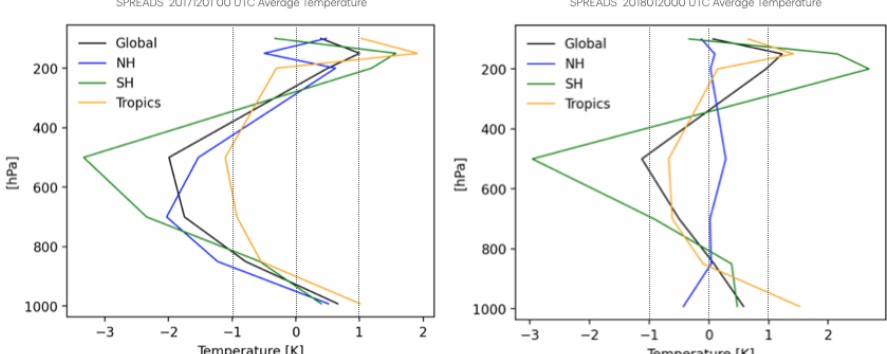

*Figure 8: Vertical profile of the average T differences between SPREADS and ERA5 for the Global (black line), NH (blue line), SH (green line) and TR (yellow line) valid at 00 UTC 20171201 (left panel) and 20180121 (right panel)*

continental regions, such as Siberia and Canada, possibly due to differences in
land-surface model physics or less dense observational constraints in those areas. The
thermal contrast between land and ocean is well maintained in both datasets, though
slightly smoother in SPREADS, again reflecting resolution effects.
To further understand these differences, Figure 8 shows vertical profiles of
average temperature differences (SPREADS minus ERA5) at 00 UTC for two key dates:
20171201 (left) and 20180120 (right), representing the beginning and end of the
evaluation period. The profiles are shown for Global (black), Northern Hemisphere
(blue), Southern Hemisphere (green), and Tropics (orange).
From early December to late January, the NH and Tropics show a marked
improvement, with temperature differences decreasing by up to ±1 K, particularly in
the mid-to-upper troposphere. This reflects both improved background constraint
and effective assimilation updates during the SPREADS evaluation period. In contrast,
the SH shows minimal change, likely due to sparser observational coverage,
underscoring the asymmetry in observing system density between hemispheres.
Notably, around 200–300 hPa, the NH and Tropics exhibit a transition from cold
to warm bias, suggesting a tropopause-level sensitivity that may be



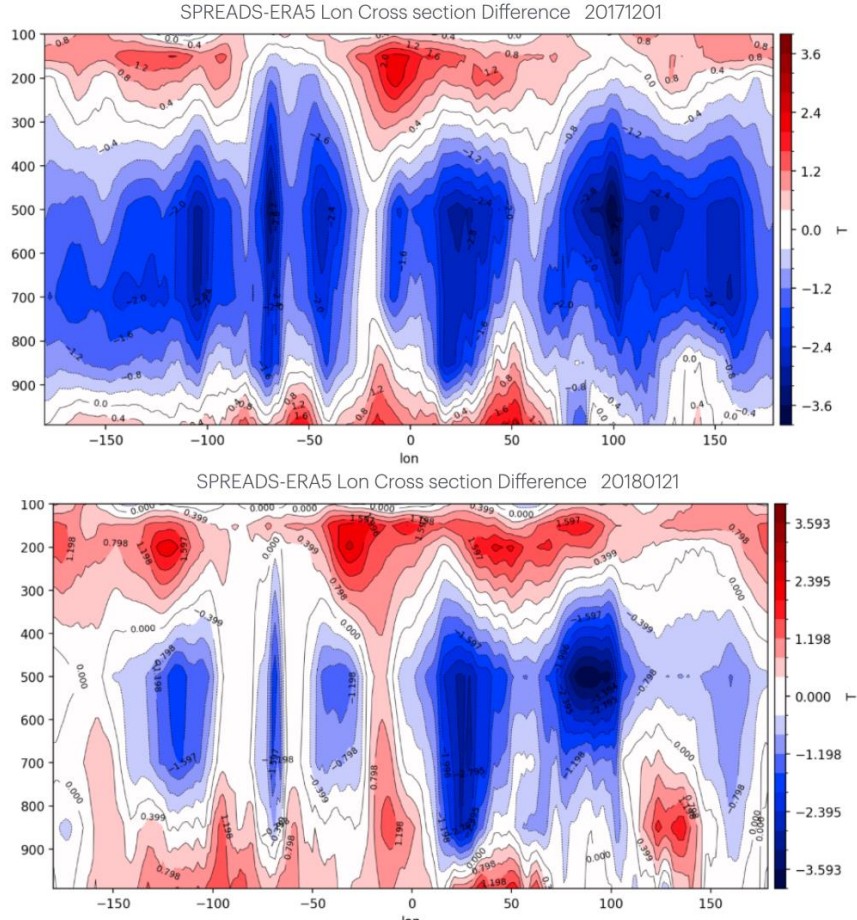

*Figure 9: Zonal cross section of the T differences between SPREADS and ERA5
valid at 00 UTC 20171201 (top panel) and 20180121 (bottom panel)*

influenced by the vertical resolution and radiative transfer modelling. Above 100,
a persistent warm bias remains globally, more pronounced in the SH, pointing to
potential limitations in stratospheric observation assimilation, possibly related to
reduced usage of GPS-RO or upper-level radiance channels. Overall, despite the
coarser resolution, SPREADS reproduces the dominant features of the atmospheric
state with high fidelity. Improvements over time in the troposphere, particularly in
the NH and Tropics, indicate that the system is maturing well. Future enhancements,
such as expanded use of satellite data in the SH, and increased spatial resolution, could
further close the gap with ERA5 in under-constrained regions.



As a final examination of the analysis difference, Fig. 9 and 10 show the average
cross section of temperature for the E-suite initial day (top panel) and at the end of
January (bottom panel)  across the longitude and latitude, respectively. At the initial
time (Fig. 9 top panel), the temperature differences between SPREADS and ERA5 show
widespread cold biases (blue shading) across nearly all longitudes in the lower to mid-
troposphere (around 900–400 hPa). Maxima of positive differences exceed +3.6 K,
particularly around the central and eastern Pacific and parts of the Atlantic. In the
upper troposphere (above ~300 hPa), some warm biases (red shading) begin to
appear, though they are less dominant. Figure 9 bottom panel shows a clear
improvement in the zonal consistency and magnitude of the differences. Cold biases
are notably reduced in amplitude and spatial extent, particularly across the mid-
troposphere. The structure becomes more vertically layered and less zonally
coherent, suggesting improved local balance and constraint. Warm anomalies aloft
become slightly more pronounced in some sectors (e.g., near 60° longitude), pointing
to evolving differences in vertical structure, possibly due to the upper-air
observational influence. The complemented Fig. 10 initially shows (top panel)
significant cold biases (~-3 to -6 K) in the extratropics, especially over the Southern
Ocean (around 60°S–80°S) and Northern Hemisphere high latitudes (~60°N–80°N),
spanning from the surface up to 400 hPa. The tropical region remains relatively
neutral, with near-zero or weakly positive anomalies. The biases show a strong
hemispheric asymmetry, being more intense and vertically extensive in the SH. At the
final time (bottom panel), the magnitude of cold biases is substantially reduced,
particularly over the Southern Hemisphere, where mid- and upper-tropospheric
differences nearly vanish. Remaining differences are more localised and patchy, with
some persistent warm
anomalies in the upper troposphere (above 300 hPa) over both poles. The tropics
remain stable, with minor variations and low-magnitude anomalies. There is an
overall flattening of the difference structure, indicating improved vertical and
hemispheric balance. In summary, between early December and late January,



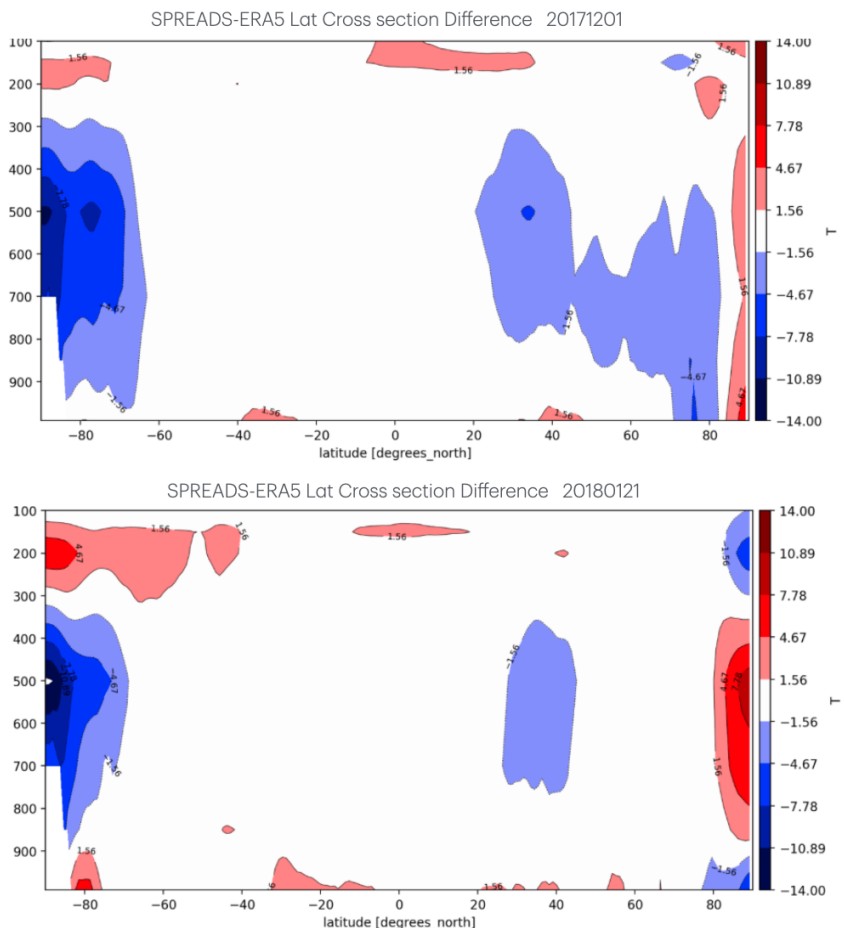

*Figure 10: Meridional cross section of the T differences between SPREADS and ERA5
valid at 00 UTC 20171201 (top panel) and 20180121 (bottom panel)*

SPREADS shows significant improvement in temperature alignment with ERA5,
particularly in the lower to mid-troposphere and across the Southern Hemisphere.
The initial large cold anomalies in the SH and NH extratropics largely diminish,
suggesting better background constraint or improved assimilation tuning over time.
There is a slight emergence of warm biases in the upper troposphere and lower
stratosphere, possibly tied to model vertical resolution (Simpson *et al.*, 2025) or under
observed stratospheric layers. In general, the differences become less globally
coherent and more structured, indicating that the system is moving toward finer-
scale, observation driven corrections rather than broad model biases.
**4. Conclusion&Plan**





The development of SPREADS, Scalable PaRallelised EArth Data Assimilation
System, represents a crucial step in advancing ensemble-based data assimilation from
research to operational application. Built upon the flexible, open-source DART
framework, SPREADS embodies the principles of transparency, collaboration, and
reproducibility that are foundational to modern Earth system science. Open-source
modelling is not only a technical choice but a strategic enabler of scientific progress:
it fosters community-driven innovation, ensures the traceability of results, and
accelerates the adoption of new ideas across institutions and research domains. By
sharing tools, code, diagnostics, and configuration options, SPREADS positions itself
at the forefront of a collaborative data assimilation system.
SPREADS introduces a suite of technical enhancements, FGAT-based temporal
alignment, modular parallelised assimilation architecture, and the d4o SQL-based
observational database to address the computational and algorithmic challenges of
operational ensemble systems. These advancements enable the efficient assimilation
of over one million observations every six hours, including a diverse set of
conventional and satellite-based measurements. The E-suite evaluation demonstrates
promising results, with improved bias characteristics, ensemble calibration, and
overall consistency in comparison to ERA5, particularly in the Northern Hemisphere
and tropics.
The bias diagnostics in SPREADS are consistent with model-based findings from
the CAM7 vertical resolution study (Simpson *et al.*, 2025). In particular, the cold biases
observed in the tropical lower stratosphere in SPREADS, as diagnosed through AMSU-
A, align with those seen in low-vertical-resolution CAM configurations. The
application of FGAT and adaptive bias correction in SPREADS significantly reduces
these biases, mirroring the improvements achieved in CAM7 through enhanced
vertical resolution. This convergence from both model and observational assimilation
perspectives underscores the robustness of the diagnostic framework in SPREADS
and its capability to detect and mitigate systematic biases in the upper troposphere
and lower stratosphere. A key direction for SPREADS development is the inclusion of
more satellite-based observations, particularly infrared radiances such as IASI. These
sensors provide rich vertical information in cloud-free conditions and are essential for
improving temperature and humidity profiles, especially in the stratosphere and
upper troposphere. Preliminary testing of IASI data within SPREADS is currently





underway and shows great promise for enhancing the vertical structure of the
analysis and addressing residual biases observed in the current system.
In parallel, all-sky microwave radiance assimilation is being actively tested. This
represents a shift in satellite data usage, enabling the assimilation of radiances under
both clear and cloudy conditions. All-sky assimilation significantly increases the
spatial and temporal coverage of radiance data, especially in regions with persistent
cloud cover such as the tropics and storm tracks. By more effectively capturing cloud-
affected observations, SPREADS aims to improve its representation of moisture fields,
cloud dynamics, and convective processes, key elements for accurate medium- to
long-range forecasts.
Although SPREADS currently operates at coarser resolution than ERA5, its ability
to replicate large-scale atmospheric patterns and to reduce biases over time
demonstrates the system's robustness. Continued tuning of satellite bias corrections,
expansion of satellite data types, and enhancement of vertical resolution will be
critical next steps. Furthermore, ongoing integration within a fully coupled Earth
system model positions SPREADS as a strategic asset for seamless forecasting, from
weather to climate timescales.
In conclusion, SPREADS is a scalable, open, and forward-looking platform that
effectively bridges research innovation with operational demands. Its modular,
transparent architecture invites community contribution and ensures adaptability to
evolving scientific goals. With a growing observational portfolio and expanding
capabilities, including the assimilation of all-sky and hyperspectral infrared radiances,
SPREADS is well positioned to become a next-generation system for global Earth
system prediction.

**Code and data availability**
All codes in this study are permanently available at
https://doi.org//10.5281/zenodo.17063454 (Cardinali *et al.*, 2025).

**Author contribution**
CC: Conceptualisation, Methodology, Supervision, Project administration,
Validation, Investigation, Writing original draft.
GC: Methodology, Code developments, Investigation



MG: Infrastructure development
SS: Observation database development and Code optimization
GD: Code developments
JA: Methodology and Code development support
KR: Software and Technical support

**Acknowledgements**

We would like to thank a few people at ECMWF who helped us make better use
of observation tools and databases, BUFR, MARS retrieval, and the archive, and who
provided insightful ERA5 maps not available online: Drasko Vasiljevic, Cristina Prates,
Tomas Kral, Manuel Fuentes, and Mohamed Dahoui. We thank Daniele Peano for
porting CAM6 to the CMCC high-performance supercomputer. A special thanks to
ESYDA Director Silvio Gualdi for his scientific vision and gentle personality, which
have supported us throughout the development and assessment of SPREADS. Many
thanks to Isla Simpson and Peter Hjort Lauritzen for providing diagnostic assessment
information on the open-source CAM6 atmospheric model (93 model levels), which is
coupled with SPREADS. Last but not least, we are very grateful to Antonio Navarra for
his vision in expanding CMCC's activities and expertise in global coupled data
assimilation.



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
