# Peer review of "SPREADS: From Research to Operational Open-Source Data Assimilation System"

_EGUsphere, 2025_

## Referee Comment (RC2)

Review for EGUsphere-2025-4294

**Specific comments:**

Figs: Nearly all figures in the manuscript are too low in resolution, and many labels, legends, and axis titles are unreadable. The figures need to be regenerated at higher quality to allow proper interpretation.

L157: Please clarify why a GUI is described as essential for running SPREADS. Most operational DA systems function without a GUI, so some justification or context would help motivate this design decision.

L203: Does the FGAT time interpolation introduce noticeable computational overhead? The text states that the approach saves CPU and memory, but the interpolation of model fields for large numbers of observations sounds potentially expensive. Some quantification or clarification would help.

L282: The test conducted by Sami Saarinen is only briefly mentioned. Please clarify whether the optimized d4o library was the sole difference between the DART and SPREADS tests. If additional differences exist, this discussion may fit better in Section 2.4 rather than Section 2.3, which focuses specifically on d4o. Also, because Saarinen is a coauthor, a personal communication citation is not needed here.

Section 3: This section would benefit from subsections to improve readability and organization. As written, it is quite verbose and does not flow well.

Fig. 3: The blue and black lines in the top panels are difficult to distinguish. Using more distinct colors would help. It would also be clearer if the radiosonde and aircraft panels used consistent y-axis scales. Additionally, I cannot locate the ensemble Dispersion Index (DI) curve. If it is the blue line in the lower panels, that should be stated explicitly. Panel labels (a, b, c, etc.) would also aid navigation.

L361: What does it mean that the RMSE is "computed against the own analysis"? Does this imply that the analysis is treated as truth? You may also note that DI is sometimes referred to as the "consistency ratio" and OI as the "fitting ratio," which may help readers familiar with alternate terminology.

L425: The statement that "posterior fits tend to converge" is unclear. The error characteristics vary substantially across observation types, so it is not obvious in what sense the fits are converging.

L433: It is unclear what is being compared in this sentence. Are you comparing column-integrated RMSE with RMSE at specific levels? The rationale for the "expected" behavior should also be clarified.

L446-458: Interpreting differences in increment spread across these two isolated dates is problematic. Many factors unrelated to system behaviour could explain these changes, including

differences in flow regime, seasonality, or observation density. I recommend removing or reframing this comparison, and I share this concern for Figures 8–10, whose panels represent different dates within a nonstationary evaluation period.

L467: The comparison between SPREADS forecasts and ERA5 would be more informative if a control experiment were included (for example, the previous operational or research DA configuration). Without this, it is difficult to quantify how much SPREADS improves upon prior capabilities. Additionally, the manuscript states that the large-scale circulation is well captured, but there are notable regional differences, such as the jet structure between northeastern Africa and India.

**Technical comments:**

L43: Recommend starting this sentence with "The". Also, this paragraph is pretty long and could be split somewhere.
L68: "DART has been used by a large world wide young and senior…": this sentence is confusing and should be rewritten.
L81: "And finally also improvements" -> "Finally, improvements"
L89: Sentence beginning "Given the crucial role…" is too long and should be cleaned up.
L148: "satellite observations, it leads" -> "satellite observations leads"
L223: Please restructure the sentence beginning "Also the scan and air mass…"
L243: "assimilation processes a query" -> "assimilation processes, a query".
Table 1: Please add "CPU time" to the table instead of "CPU".
Fig. 3 caption: Missing closing parenthesis after (OI). Also, the spread calculations should not be considered as a time series of o-b or o-a departures.
L359: The RMSE being larger for aircraft
Fig. 6: What does the (0.3) mean in the legend here? Also, please improve caption. What does each dot represent in the right column?
L526: Accidental paragraph break?